# Universal wing- and fin-beat frequency scaling

**Jens Højgaard Jensen, Jeppe C. Dyre, Tina Hecksher***

Department of Science and Environment, Roskilde University, Roskilde, Denmark

* tihe@ruc.dk

## Abstract

We derive an equation that applies for the wing-beat frequency of flying animals and to the fin-stroke frequency of diving animals like penguins and whales. The equation states that the wing/fin-beat frequency is proportional to the square root of the animal's mass divided by the wing area. Data for birds, insects, bats, and even a robotic bird—supplemented by data for whales and penguins that must swim to stay submerged—show that the constant of proportionality is to a good approximation the same across all species; thus the equation is universal. The wing/fin-beat frequency equation is derived by dimensional analysis, which is a standard method of reasoning in physics. We finally demonstrate that a mathematically even simpler expression without the animal mass does not apply.

**Data Availability Statement:** All data used in the present paper are collected from published papers. For convenience we have tabulated all data in the Supporting information with appropriate references.

## Introduction

Convergent evolution occurs when different species occupying similar ecological niches adapt in a similar way to similar selective pressure [1]. The evolution of the capacity of flight is a classic example of this. Flying insects, birds, and bats—but also the prehistoric reptiles, pterosaurs—all evolved the useful ability to fly. The present work shows how insight into the wing-beat frequency may be arrived at from arguments solely based on so-called dimensional analysis [2–6], which expresses the requirement that the physical dimensions must be the same on both sides of an equality sign. Physics uses "quantity calculus" in contrast to the "number calculus" of pure mathematics [3]. To illustrate the difference and what a quantity is, consider the wing area $A$. If $A$ is 100 cm$^2$, it is also 0.01 m$^2$ or 0.108 ft$^2$. The area $A$ may thus be expressed by different products of numbers and units. The units all refer to the quantity "area", which has the dimension "length squared". Numbers and units are needed when comparing theoretically derived formulas to measurements, whereas one usually derives theoretical results for physical quantities without reference to units. Adding, subtracting, and equating two quantities is only allowed if these have the same dimensions; thus adding quantities of different dimensions is meaningless. New physical quantities with derived dimensions can be created, however, by multiplying or dividing physical quantities (e.g., velocity = length/time).

The use of dimensional analysis and other general physics arguments has a long history for estimating the wing-beat frequency $f$ of flying animals [7–16], as well as for deriving scaling relations between different morphological parameters of flying and swimming animals [17–24]. This has led to a number of mathematically sophisticated expressions. For instance, Pennycuick [12] arrived at $f = K\, m^{1/3} g^{1/2} S^{-1} A^{-1/4} \rho_{\mathrm{air}}^{-1/3}$ in which $K$ is a numerical constant by

**Funding:** The work was supported by the VILLUM Foundation's Matter grant VIL16515. The funders had no role in study design, data collection and analysis, decision to publish, or preparation of the manuscript.

**Competing interests:** The authors have declared that no competing interests exist.

finding the best fit to a set of bird data involving the following parameters: the bird mass $m$, the gravitational acceleration $g$, the wing span $S$, the wing area $A$, and the air density $\rho_{air}$. Puranik *et al.* [25] suggested $f \propto mL^{-2}B_{eff}$, where $L$ is the wing span and $B_{eff}$ the effective wing breath defined as wing area divided by wing length, and found a good fit to data for both insects, birds, and bats. More recently, mathematically simpler expressions, e.g., a proportionality between mass to some power and wing/stroke frequency, have been considered [14, 22, 24, 26]. Most expressions discussed in the literature have focused on making predictions for a specific group of flying animals, for instance insects [7, 16, 27] (or even specific insect orders [21]) or birds [12, 13, 15, 28], and for good reasons: There is obviously a significant difference between the flight of, e.g., a dragonfly and that of a bat.

To motivate our analysis, we show in Fig 1 literature data for very different flying animals. Large animals are expected to flap their wings more slowly, so it is not surprising that there is a clear correlation between animal mass and wing-beat frequency (Fig 1(a)). An even stronger correlation is observed between the mass of the animal and the wing area, a correlation that expresses the obvious fact that large animals have large wings (Fig 1(b)).

As a first, pragmatic attempt into finding a general relation between wing-beat frequency and morphological parameters, Fig 1(c) and 1(d) show log-log plots of the wing-beat frequency as a function of inverse wing area, $1/A$, and inverse mass, $1/m$, respectively. The best-fit slopes in these plots are 0.42 and 0.30, corresponding to the power laws $f \propto A^{-0.42}$ and $f \propto m^{-0.30}$. In both cases the correlations are convincing. However, these power-law dependencies are purely empirical and cannot be derived by simple arguments. The question we address in this paper

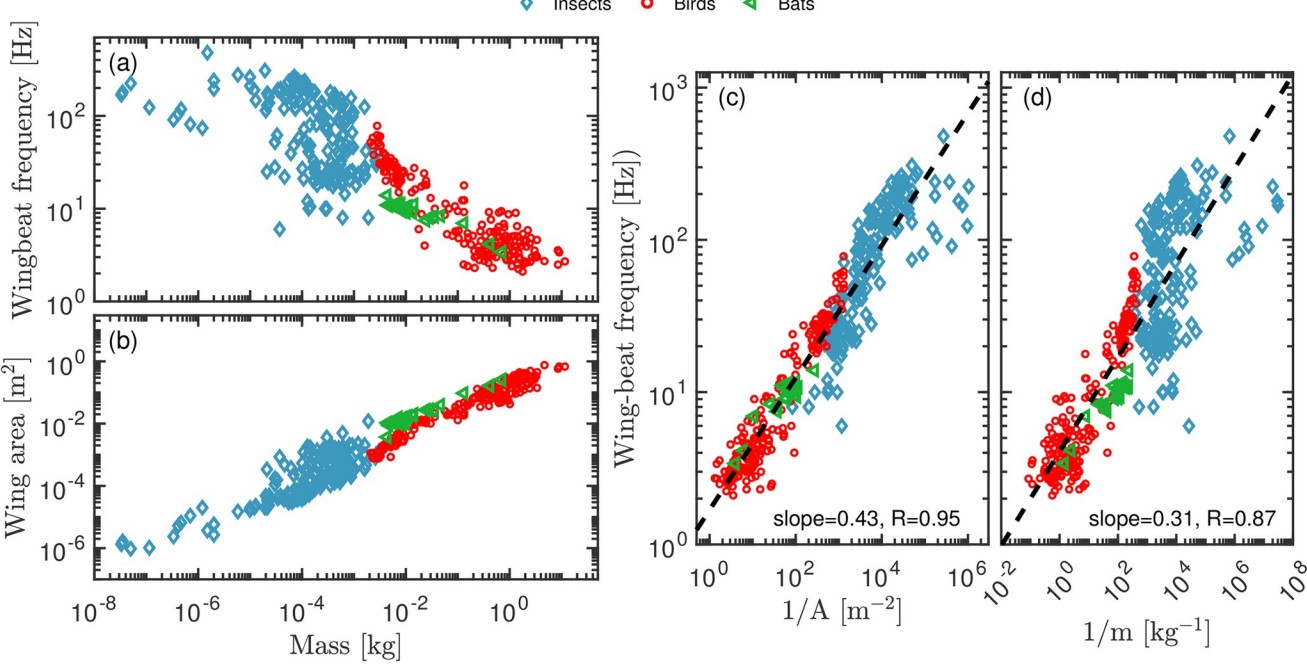

**Fig 1. Correlations between wing area, wing-beat frequency, and mass of flying animals.** Literature data for insects (blue diamonds) [21, 29–31], birds (red circles) [12, 13, 15, 30, 32, 33], and bats (green triangles) [32, 34]. (a) Wing-beat frequency versus mass; (b) wing area versus mass. These data confirm the intuition that large flying animals have a low wing-beat frequency and large wings. (c+d) Tests of power-law expressions for the wing-beat frequency (same data as in a+b): (c) $f$ versus $1/A$; (d) $f$ versus $1/m$. In both cases significant correlations are observed, with the Pearson correlation coefficients 0.95 and 0.85, respectively, and vanishing $p$-values... The best-fit slopes are 0.42 and 0.30, corresponding to power-law expressions $f \propto m^{-0.43}$ and $f \propto A^{-0.31}$. These empirical findings cannot be justified from first principles, however.

is: can one arrive at a reliable prediction for the wing-beat frequency of a flying animal from physics-based arguments alone, i.e., without appealing to empirical correlations?

## A theoretically derived simple expression for the wing-beat frequency of hovering animals

It is a tremendous challenge to account fully for the fluid dynamics of wing flight, because it involves the force due to wing flapping as well as the lift due to the circulation around the wings that depends on details of the flow vortex structure [35]. These two forces jointly compensate for gravity and drag during steady-state horizontal flight. To take a simple approach to the problem, we consider the case of a hovering animal.

A hovering animal must create an upward force, $F_{up}$, to balance the downward gravitational pull, i.e.,

$$F_{up} = mg\,. \tag{1}$$

Force balancing is obtained by the wings pushing air downwards, which according to Newton's third law creates a lift. Newton's second law implies that the lift force is equal to the downward air momentum per time created by the wings. This momentum—and consequently $F_{up}$—varies during the wing stroke. Both $F_{up}$ and the created momentum per time (and all other quantities) are in the following taken to be averages over one wing stroke. The created downward momentum of the air per unit of time is the product of the air mass set in motion per time and its average velocity, $v_{air}$. Thus Newton's second law implies

$$F_{up} = \frac{\Delta m_{air}}{\Delta t} v_{air}\,. \tag{2}$$

Here $\Delta m_{air}$ is the mass of air pushed down in a stroke and $\Delta t$ the time of one stroke. The mass of air set in motion per time is the density of air, $\rho_{air}$, times the volume of air set in motion per time. This volume is the cross sectional area of the downwards air flow, $A_c$, times the downward air velocity. We thus arrive at

$$F_{up} = \rho_{air} A_c v_{air}^2\,. \tag{3}$$

This expression is difficult to test as it contains variables, $A_c$ and $v_{air}$, that are not easily measured. Instead we determine how $F_{up}$ depends on the wing area $A$ and the wing-beat frequency. For this, dimensional analysis is invoked.

A wing stroke is characterized by many more kinematic quantities than the wing area and the wing-beat frequency, for instance the wing-amplitude angle, ratios of lengths and angles describing the shape of the wings, ratios of time intervals describing details of the wing motion. These quantities all have an effect on the wing beat frequency [23, 27, 36, 37]. Angles and ratios of quantities of the same kind are all dimensionless, however. For dimensional reasons, the area $A_c$ of Eq 3 must therefore be equal to the wing area $A$ times a dimensionless function of all dimensionless quantities describing the shape of the wings. Similarly, $v_{air}$ can be written as the product of wing span and wing-beat frequency times a dimensionless function of the dimensionless quantities describing details of the shape and movements of the wings, and the wing span can be expressed as $A^{1/2}$ times a dimensionless function of dimensionless quantities describing the shape of the wings. Thus, $v_{air}$ is $A^{1/2}f$ times a dimensionless function of dimensionless quantities describing the shape and movement of the wings. Inserting the expressions for $A_c$ and $v_{air}$ into Eq 3, we thus get

$$F_{up} = C\rho_{air}A^2 f^2 \tag{4}$$

in which $C$ is a combination of all the unknown dimensionless functions of dimensionless quantities entering into the problem. Combining Eqs 1 and 4 one arrives at the following expression for the wing-beat frequency of a hovering animal

$$f = \sqrt{\frac{mg}{C\rho_{\text{air}}A^2}}.$$ (5)

If one from the beginning were convinced that $F_{\text{up}}$ of a flying animal is uniquely determined by $\rho_{\text{air}}$, $A$, and $f$, then dimensional analysis may be applied directly to derive Eq 5. To see this, we write $F_{\text{up}} = C\rho_{\text{air}}^{\alpha}A^{\beta}f^{\gamma}$ and ask for the values of $\alpha$, $\beta$ and $\gamma$ securing same dimensions on both sides of the equation. Taking length, time, and mass as basic dimensions, the dimension of the force $F_{\text{up}}$ is mass × length × (time)$^{-2}$, while $\rho_{\text{air}}$, $A$, and $f$ have dimension mass × (length)$^{-3}$, (length)$^2$, and (time)$^{-1}$, respectively. In order to get the mass dimension right on both sides of the equality sign in $F_{\text{up}} = C\rho_{\text{air}}^{\alpha}A^{\beta}f^{\gamma}$, one must have $\alpha = 1$ since neither $A$ nor $f$ contain the dimension mass. Similarly, $\gamma = 2$ in order to get the time dimension right. Finally, $2\beta - 3\alpha = 2\beta - 3 = 1$ to get the correct length dimension, which leads to $\beta = 2$. We thus arrive at Eq 5.

If $C$ is the same dimensionless function value for all animals, we obtain by ignoring the minor variations of air density and gravitational field strength the following simple proportional relationship for the wing-beat frequency of a hovering animal [38]

$$f \propto \frac{\sqrt{m}}{A}.$$ (6)

The functional form of Eq 6 follows from laws of physics. This mean that other power-law functions discussed in Ref. [27], such as Norbergs $f \propto m^{0.3}$ [14], Pennyquick's $f \propto m^{1/3}$ [12] or $f \propto m^{3/8}$ [13], or the mass-flow theory arriving at $f \propto m$ [25], cannot be derived from purely physical arguments.

In 1970 Deakin arrived at Eq 6 for insect flight by a dimensional analysis involving the five variables $\rho_{\text{air}}$, $m$, $g$, $A$, and $f$ [7]. Deakin applied the Buckingham pi theorem [2], which implies $\rho_{\text{air}}A^{3/2}/m = C_1$ and $f^2A^{1/2}/g = C_2$ where $C_1$ and $C_2$ are dimensionless, numerical constants. To arrive at Eqs 5 and 6 Deakin utilized the fact that it is found experimentally that $f \propto 1/A$ applies to a good approximation for insects of same mass. This means that one can write $f = \phi(m)/A$ for some function $\phi(m)$, which implies $\phi^2(m)A^{-3/2}/g = C_2$ or $A^{3/2} = \phi^2(m)/(C_2g)$. Substituted into the $C_1$ Buckingham equation, this results in $\rho_{\text{air}}\phi^2(m)/(mg) = C_1C_2$. Combining this with the experimental observation $f = \phi(m)/A$ one arrives at Eq 5 which, as mentioned, reduces to Eq 6 if the minor variations of air density and gravitational field strength are ignored.—The difference between Deakin's argument and the above presented physical reasoning is that the latter makes use of the fact from physics that $m$ and $g$ must enter into the problem in terms of the single quantity "weight", $mg$. This reduces the number of variables from five to four, implying that $f$ is determined by just three input variables, in which case Eq 5 follows without empirical input.

Interestingly, Eq 5 also gives a recipe for the fin/fluke frequency of swimming animals as positively buoyant diving animals must continuously move water upwards in order to stay submerged. This, of course, holds only for animals with no means of adjusting buoyancy, which excludes fish with a swim bladder [39]. Calculating fin/fluke frequency from Eq 5 requires two modifications, replacing $\rho_{\text{air}}$ by $\rho_{\text{water}}$ and $m$ by the buoyancy-corrected mass, $m(\rho_{\text{water}} - \rho_{\text{animal}})/\rho_{\text{animal}}$. Thus using Eq 6 to compare data for swimming/diving animals in the same plot as flying animals, a correction factor accounting for the difference in water and air densities, $\sqrt{\rho_{\text{air}}/\rho_{\text{water}}}$ is needed for the data points for swimming/diving animals.

## Results and discussion

Eq 6 is tested in Fig 2 for the same data as in Fig 1 by plotting the wing-beat frequency versus $\sqrt{m}/A$. The idea is to make use of the fact that the flight wing-beat frequency is close to that of hovering [7, 12]. The argument is that—like the frequency of walking—the wing-beat frequency is largely determined physiologically as the natural (resonance) frequency of the wing, where the energetic cost of the flapping movement is minimized. The wing-beat frequency may not always be exactly at resonance for animal-specific reasons (see e.g. [40]), but it will not deviate significantly (an eagle will never flap its wings at the frequency of a fly and vise versa) and, notably, whatever specific reason for a deviation from resonance must hold for hovering and forward flight alike.

The data set consists of a total of 414 data points for flying animals reported in the literature. They are distributed on classes as follows: 176 different insect data points [21, 30, 31] (including bees, moths, dragonflies, beetles, and mosquitoes), 212 different bird data points [12, 13, 15, 17, 18, 32, 33, 41] (ranging from hummingbirds to swans), and 25 bat data points [32, 34]. Most data points are based on measurements/observation of a single individual animal, some on a few different individuals in which case the numbers are averages. The figure also includes an ornithopter (artificial bird) data point [42], conforming to the general pattern of animals.

The uncorrected data for swimming/diving animals are included as open squares and triangle down symbols in Fig 1. To make a proper comparison to flying animals, these data points were corrected according to the procedure described above with the values of $\rho_{air}$ = 0.0012 g/cm$^3$ and $\rho_{water}$ = 1.0025 g/cm$^3$ and are shown with the same filled symbols.

We did not find any publications with all the required information: fluke/fin area, fluke/fin frequency, and two of the following three: animal mass, animal density, animal volume. Hence, to make the comparison we pieced together data from different publications and in some cases estimated the animal density based on other information (e.g., body composition). For deep-diving animals, the density increases with diving depth due to compressing, especially of air-filled organs such as lungs and air trapped in feathers; thus same-depth values for all of the quantities would be optimal, but unfortunately these are rarely available.

We included data points for penguins [24] with a density of 0.9 g/cm$^3$ at the surface (estimated from Eq. 3 in Ref. [45]). Data points for large whales are from Ref. [43] with an estimated density of 1.034 g/cm$^3$ as an average of the available data: Narazaki *et al* [46] showed that humpback whales are close to neutrally or negatively buoyant with densities between 1.025 to 1.043 g/cm$^3$, and nearly the same is shown for northern bottlenose whale [47]). A series of data points for smaller whales [44] were also included with an estimated density of 0.95 g/cm$^3$. Small whales have a relatively larger body mass percentage of blubber compared to larger whales. The fin/fluke beat frequency of swimming animals vary more than for flying animals. The frequency used in our study is always an average of frequencies stated in a given reference if more than one number is given. Although the data points of swimming/diving animals come with a larger uncertainty due to these corrections and estimates, they nevertheless conform well to the same line as the flying animals (Fig 2). Other interesting classes of swimming/diving animals include pinnipeds and batoids; however, we did not find any (even partially) complete data sets in the literature on these animals.

The data support a version of Eq 6 in which the constant of proportionality varies little between different classes of animals. This is striking in view of the fact that the dimensionless function $C$ in Eq 5 could well be completely different for different animals. As the wing shapes of, e.g., a butterfly and a bat are by no means similar, it appears that evolution somehow has

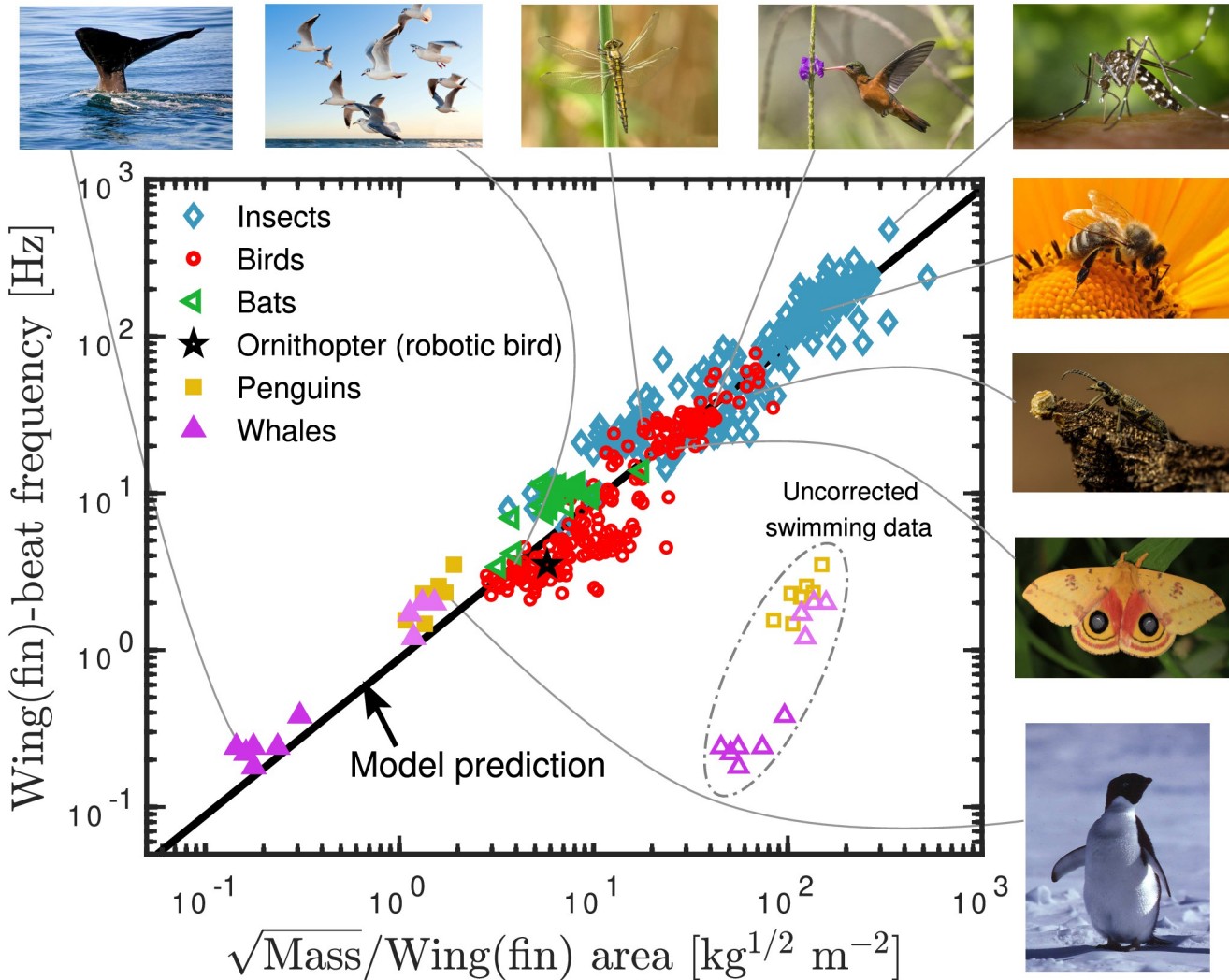

**Fig 2. Wing-beat-frequency data for a variety of flying animals versus the square-root of the animal mass divided by the wing/fin area (same data as in Fig 1).** There is some scatter in the data, which is not surprising given that these are for quite different animals, but to a good approximation the data fall on the same line. The model prediction is the full black line of unity slope unity and the Pearson correlation coefficient is 0.95. Data for swimming animals, corrected for buoyancy and the difference in water and air density (see main text for details), have been included in the plot. The uncorrected data for swimming penguins (open squares) [24], large whales [43] (open purple triangles), and small whales [44] (open light purple triangles) clearly do not conform to the general trend for flying animals. However, with the appropriate corrections, the data points (filled symbols) fall on the continuation of the best fitted model prediction line of the flying animals. Photo attributions: Hummingbird, republished from wikimedia with no changes made under a CC BY license (https://creativecommons.org/licenses/by-sa/4.0/deed.en), with permission from Charles J. Sharp: https://commons.wikimedia.org/wiki/File: Cinnamon_hummingbird_(Amazilia_rutila)_in_flight_Los_Tarrales.jpg. Moth, republished from Flickr under a CC BY license (https://creativecommons. org/licenses/by/2.0/), with permission from Judy Gallagher: https://www.flickr.com/photos/52450054@N04/35499910606. Penguin, republished from Hans Ramløv under a CC BY license, with permission from Hans Ramløv, original copyright [1991]. Beetle, republished from Thorbjørn Ramløv under a CC BY license, with permission from Hans Ramløv, original copyright [2018]. Whale, retrieved from RawPixel at https://www.rawpixel.com/image/4022862/ whale-originalpublic-domain-image-from-flickr. Seagulls, retrieved from Pexels at https://www.pexels.com/photo/white-seagulls-flying-over-the-ocean-54462/. Mosquito, retrieved from Pexels at https://www.pexels.com/photo/black-white-mosquito-86722/. Bumble bee, retrieved from Pexels at https://www. pexels.com/photo/bumble-bee-on-yellow-daisy-67560/. Dragonfly, retrieved from Pexels at https://www.pexels.com/photo/yellow-and-black-dragonfly-on-green-stem-during-daytime-56010/.

adjusted the dimensionless arguments going into $C$ to give roughly the same value of $C$ across all species.

As mentioned, dimensional analysis cannot give the value of the dimensionless constant $C$ in Eq 5, which must be determined empirically. We find the constant of proportionality in Eq

[6](i.e., $\sqrt{g/C\rho_{\text{air}}}$) to be 0.88 $\text{m}^2/(\text{s}\sqrt{\text{kg}})$ from which one obtains $C = g/(\rho_{\text{air}} \cdot (0.88\text{m}^2/(\text{s}\sqrt{\text{kg}}))^2) \simeq 11.$

As an application of Eq 6 with an almost universal value of $C$, we estimate the wing-beat frequency of the largest known flying animal, the extinct pterosaur Quetzalcoatlus northropi. With an estimated weight of 65 kg and an estimated wing area of 10 $\text{m}^2$ [19, 48], the Quetzalcoatlus northropi predicted wing-beat frequency is 0.7 Hz.

We have shown that Eq 6 applies for animals of very different sizes and shapes. Is there no limit to the range of validity of this expression? The answer is yes, because at sufficiently small Reynolds number Re, different physics come into play [36, 49]. The animals included in Fig 2 cover the range $10 < \text{Re} < 1,000,000$. At such high Reynolds numbers the interaction between a moving body and a fluid is dominated by momentum transfer deriving from the body pushing the fluid aside; in this case the fluid density is important, while the viscosity is not. At much lower Reynolds number the body-fluid interaction primarily takes place by momentum diffusion, which according to the Navier-Stokes equation is controlled by the viscosity $\eta$, while the fluid density is irrelevant. In the first case energy is transferred from body to fluid in the form of macroscopic kinetic energy, in the second case in the form of heat (microscopic kinetic energy). Very low Reynolds numbers correspond to flying animals that are much smaller than existing ones. Having in mind, e.g., a flying nanorobot, however, one may nevertheless ask: What is the prediction for the wing-beat frequency when $\text{Re} \ll 1$? To answer this, dimensional analysis may again be applied, this time assuming that the lift force generated by the wings to balance the gravitational pull is a function of $\eta$, $A$, and $f$. Recalling that the dimension of viscosity is mass $\times$ (length)$^{-1} \times$ (time)$^{-1}$, the result is $f \propto mg/(\eta A)$. Thus for flying animals at very small Reynolds numbers, $f \propto m/A$ replaces the above $f \propto \sqrt{m}/A$.

Given the success of Eq 6 with a universal proportionality constant, the obvious question is: Can data be described by an even simpler expression? If it is assumed that the shapes of all flying animals are similar, the wing area is determined by a characteristic length $L$ squared, $A \propto L^2$, and the animal volume $V$ by the same length cubed, $V \propto L^3$. The mass of the animal is given as $m = \rho_{\text{animal}}V$ in which $\rho_{\text{animal}}$ is the animal's density. Plugging these expressions into the wing-beat formula Eq 6 we obtain

$$f \propto \frac{\sqrt{m}}{A} \propto \frac{\sqrt{\rho_{\text{animal}}L^3}}{L^2} = \sqrt{\frac{\rho_{\text{animal}}}{L}}. \tag{7}$$

Since the animal densities do not vary by orders of magnitude, we see that the simple proportionality $f \propto 1/\sqrt{L}$ applies under the assumption of similar shapes. An obvious choice of $L$ is the square root of the wing area $A^{1/2}$, and thus the proportionality to be tested is $f \propto 1/A^{1/4}$.

Fig 3 plots all data of Fig 2 as a function of the inverse of the square root of the characteristic length. Under the assumption of similar shapes these data points should fall on a line with slope 1 (full line). Instead the data follow a different trend with the best fit slope of 1.68 (dashed line) (this is consistent with the found slope in Fig 1(c)).

## Summary

We have derived a simple equation for the wing-beat frequency of hovering animals from the laws of physics. We tested this equation toward empirical data by making use of the fact that the wing-beat frequency of hovering animals and forward flying animals is roughly the same [7, 12]. A comparison to empirical data shows that Eq 6 holds well for all flying animals, with a constant of proportionality that is nearly the same for all species. Swimming animals conform to the same striking approximate universality if one accounts for buoyancy and the air/water

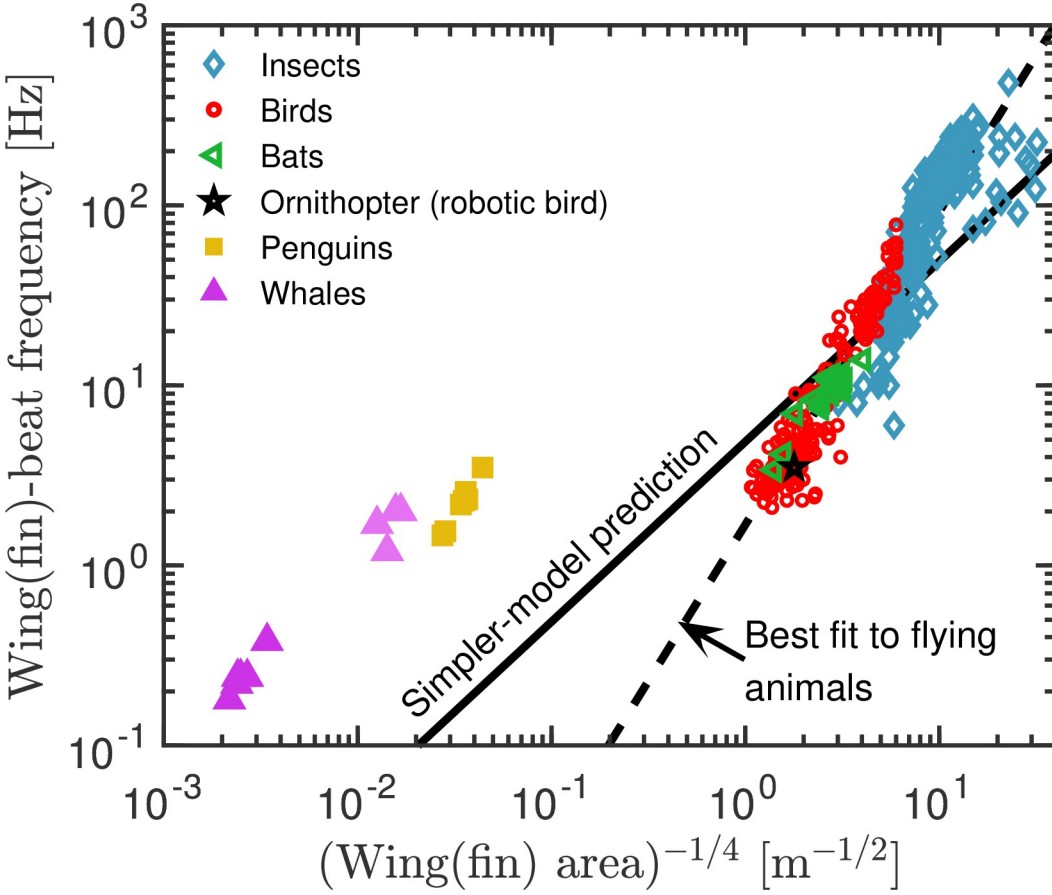

**Fig 3. Testing a simpler expression for the wing-beat frequency data (same data in Fig 1).** The plot shows a test of Eq (7), which is derived from Eq (6) assuming similar shapes and movements for all flying animals. The data do not follow the predicted proportionality $f \propto A^{-1/4}$ indicated by the full line. Instead, the best fit (dashed line) yield $f \propto (A^{-1/4})^{1.68} = A^{-0.42}$, which is consistent with Fig 1(c).

difference in density. Finally, we demonstrated that the wing-beat frequency equation cannot be simplified further.

## Supporting information

**S1 File. The supporting information for this paper contains the following: i) Licence status and links to the images used in Fig 2 and ii) the full set of data points used for Figs 1–3 tabulated with appropriate references.**
(PDF)

## Acknowledgments

As physicists we are grateful to biologists Eline Lorenzen, Hans Ramløv, Thomas Kiørboe, and Tobias Wang for help and encouragement.

## Author Contributions

**Conceptualization:** Jens Højgaard Jensen, Jeppe C. Dyre, Tina Hecksher.

**Data curation:** Tina Hecksher.

**Formal analysis:** Jens Højgaard Jensen, Tina Hecksher.

**Funding acquisition:** Jeppe C. Dyre.

**Investigation:** Tina Hecksher.

**Methodology:** Jens Højgaard Jensen, Tina Hecksher.

**Writing – original draft:** Jens Højgaard Jensen, Jeppe C. Dyre, Tina Hecksher.

**Writing – review & editing:** Jens Højgaard Jensen, Jeppe C. Dyre, Tina Hecksher.

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
