## [Decision Letter · Decision Letter 0]

6 Jul 2023

PONE-D-23-07976Universal wing- and fin-beat frequency scalingPLOS ONE

Dear Dr. Hecksher,

Thank you for submitting your manuscript to PLOS ONE. After careful consideration, we feel that it has merit but does not fully meet PLOS ONE’s publication criteria as it currently stands. Therefore, we invite you to submit a revised version of the manuscript that addresses the points raised during the review process.

We look forward to receiving your revised manuscript.

Kind regards,

Roi Gurka

Academic Editor

PLOS ONE

“This work was supported by the VILLUM Foundation’s Matter grant (No. 16515).”

“The work was supported by the VILLUM Foundation's Matter grant (No. 16515) as mentioned in the acknowledgments. The funders had no role in study design, data collection and analysis, decision to publish, or preparation of the manuscript.”

4. We note that Figure 2 in your submission contain copyrighted images. All PLOS content is published under the Creative Commons Attribution License (CC BY 4.0), which means that the manuscript, images, and Supporting Information files will be freely available online, and any third party is permitted to access, download, copy, distribute, and use these materials in any way, even commercially, with proper attribution. For more information, see our copyright guidelines: http://journals.plos.org/plosone/s/licenses-and-copyright.

1. You may seek permission from the original copyright holder of Figure 2 to publish the content specifically under the CC BY 4.0 license.  

Reviewers' comments:

Reviewer's Responses to Questions

**Comments to the Author**

1. Is the manuscript technically sound, and do the data support the conclusions?

Reviewer #1: No

Reviewer #2: Partly

2. Has the statistical analysis been performed appropriately and rigorously? 

Reviewer #1: I Don't Know

Reviewer #2: Yes

3. Have the authors made all data underlying the findings in their manuscript fully available?

Reviewer #1: No

Reviewer #2: Yes

4. Is the manuscript presented in an intelligible fashion and written in standard English?

Reviewer #1: No

Reviewer #2: Yes

5. Review Comments to the Author

Reviewer #1: The manuscript explores whether there is a universal law that explains how wing beat frequency scales with wing area and body mass in flying and swimming animals. A power law equation is proposed, and it was validated based on the experimental data provided in the literature. There are three major areas that need improvement before I recommend it for publication in PLOS.

1) Quality of writing

- Some sections of the manuscript reads like text book (e.g., first paragraph of the introduction, first part of the Materials and methods section)

- There are many speculative sentences with no justification or

- The description of the collated data from the literature is missing. Without more information of the data, it is difficult to judge the validity of the assumptions made (e.g., hovering assumption).

- The parameters should be defined when they were first introduced. For instance, the parameters of the first equation in the second paragraph of the introduction were not defined.

- Some technical terms are used very "loosely".

- More details should be given about how the scaling law was derived considering fluid and internal forces of the animals.

2) Justification of the research

What are the research objectives, novelty compared to previous research and impact of the main findings? From a naive reader perspective, I don't think there is not enough context to appreciate the study.

3) Methodology

I find simplifying the problem to "hovering behaviour" unjustified. Similarly , the assumptions of "natural frequency" of wing beats and "gravity mediated fin movements in swimming animals" (without considering wing amplitude, Reynolds number especially comparing at animals at different scales, etc.) are too simplistic and unrealistic. In addition, the evaluation of the proposed equation using r-square value is also not very informative given that the relationship between wing beat frequency, wing surface area and mass has already been established in previous studies. I think reporting results individually one for each species is important to evaluate the generalisability of the model (including number of animals and number of n for each animal).

Reviewer #2: A universal formula that relates flapping frequency to morphological parameters for propulsive locomotion through air and water is an important scientific contribution. In this paper, the authors derive Deakin’s formula (that relates wingbeat frequency, body mass and wing area) by dimensional analysis and using simplifying physical assumptions. They present that the formula applies across flying animals as well as some swimming and diving animals in the context of their fin-beat frequency, once corrections in medium density and force of buoyancy are applied. Overall, the paper is well-written, and most of the work is rigorously explained. However, the paper requires some major revisions before being published which are proposed as follows.

The swimming animal data points presented in the paper are from one blue whale and a few penguins. A quick observation is that the authors do not mention the source of the blue whale data point. Nevertheless, it is an interesting observation that the data points from these swimming animals follow the same linear relationship as flying animals once the density corrections suggested by the authors are applied. However, the relationship presented for swimming animals will be impactful if it generalizes across not just penguins but also other diving animals including pinnipeds and cetaceans as well as fin-flapping fish such as batoids. Therefore, I recommend that some data from these groups of swimming animals should also be included in the paper. In case, there is a serious challenge in acquiring or presenting this data, this should be elaborated.

An important contribution presented in the paper is the two correction factors that account for air and water density differences. It would be useful if the authors also include the exact numerical details of how the data points from diving animals were manipulated through the correction factors before plotting them for linear regression in Fig. 2. In addition, it would also be useful to see how these data points look relative to the linear regression plot without applying the proposed correction factors. This would show the magnitude of the effect of the proposed correction factors.

Another important contribution of the paper is that the authors derive Deakin’s wingbeat frequency scaling equation from “less general” arguments using conditions for Newton’s third law at hovering steady-state. I think it would improve the paper’s impact if the authors explain why Deakin’s arguments were less general than theirs. In addition, the inclusion of a discussion of chapter 6 of “Biophysics of Insect Flight” (by N. Chari, P. Mukkavilli, L. Parayitam published by Springer) would also be useful because this chapter makes similar arguments as those used by the authors in this study. Nevertheless, it is interesting that despite the simplifying physical assumptions, the equation extends to non-hovering data including forward flying and diving animals. It is critical that the authors provide some physical intuition on how a relationship derived using the hovering assumption can generalize for non-hovering situations besides the argument that wing shape and movement do not have a big influence or the fact that underlying laws of physics are the same. For instance, any insights using similar steady-state conditions during hovering, forward flight and swimming can be useful. Moreover, the data used or presented for wingbeat frequencies for flying animals is well-placed in the context that it is the flapping frequency close to that of hovering, or in other words, sufficient to balance the weight of the animal. Because the idea of the wingbeat frequency formula is being extended for the swimming animals, with the observation that fin-beat frequency during swimming can vary more than wingbeat frequency during flight, a natural question arises here: What value of fin-beat frequency is to be used in a comparable context? Is it the maximum value of an individual, an averaged representative value, or a value that sufficiently balances the forces on an animal underwater?

The authors conclude that the wingbeat frequency formula with a universal proportionality constant works across species to a remarkable degree. To improve the utility of the information presented in this paper, the precise value of this proportionality constant must be given. Even though the authors give an example of the predictive capability of the formula, a detailed discussion on a wider scope of its predictive capability (for example in phylogenetic and allometric studies) would also improve the paper’s impact. An important parameter in the analysis is the dimensional number C in equations 2 and 3 which incorporates the combined effects of the animal’s wing shape and wing movement (excluding wingbeat frequency). In lines 128-129, the authors say that C generalizes because animals have similar wing shapes and movements. This seems to be an oversimplification because there can be large variations in wing shapes and movements even within clades, particularly insects. A more appropriate conclusion can be that the effects of wing shapes and movements combine to have roughly similar effects across animals.

6. PLOS authors have the option to publish the peer review history of their article (what does this mean?). If published, this will include your full peer review and any attached files.

Reviewer #1: No

Reviewer #2: **Yes: **Usama Bin Sikandar

---

## [Author Response · Author response to Decision Letter 0]

22 Feb 2024

All comments from the editor is addressed in the cover letter and response to reviewer comments are uploaded as a pdf-file under "Attach files".

---

## [Decision Letter · Decision Letter 1]

9 Apr 2024

PONE-D-23-07976R1Universal wing- and fin-beat frequency scalingPLOS ONE

Dear Dr. Hecksher,

Thank you for submitting your manuscript to PLOS ONE. After careful consideration, we feel that it has merit but does not fully meet PLOS ONE’s publication criteria as it currently stands. Therefore, we invite you to submit a revised version of the manuscript that addresses the points raised during the review process.

We look forward to receiving your revised manuscript.

Kind regards,

Roi Gurka

Academic Editor

PLOS ONE

Journal Requirements:

Reviewers' comments:

Reviewer's Responses to Questions

**Comments to the Author**

1. If the authors have adequately addressed your comments raised in a previous round of review and you feel that this manuscript is now acceptable for publication, you may indicate that here to bypass the “Comments to the Author” section, enter your conflict of interest statement in the “Confidential to Editor” section, and submit your "Accept" recommendation.

Reviewer #2: (No Response)

2. Is the manuscript technically sound, and do the data support the conclusions?

Reviewer #2: Yes

3. Has the statistical analysis been performed appropriately and rigorously? 

Reviewer #2: Yes

4. Have the authors made all data underlying the findings in their manuscript fully available?

Reviewer #2: (No Response)

5. Is the manuscript presented in an intelligible fashion and written in standard English?

Reviewer #2: Yes

6. Review Comments to the Author

Reviewer #2: The authors have made some significant improvements to the manuscript and addressed most of the reviewer comments. Justifications have also been provided in cases where the reviewer comments could not be fully addressed. However, I recommend following minor revisions before accepting the article.

• Even though the correlations look significant but please provide p values for your correlations. For instance, in Fig. 1.

• In Fig. 1 caption, I think the authors meant “blue diamonds” and “red circles”.

• Lines 69-71: It would be helpful for the reader if authors could specify whether by characterizing a wing stroke they mean kinematics, dynamics or both. Please also cite some literature when mentioning the effect of each of these parameters on the wing stroke.

• Line 99: The authors define the proportionality symbol here but they previously used it in line 40 and in Fig.1 caption. I’d suggest removing this line and rather replacing “simple expression” in lines 97-98 with “simple proportional relationship” or just “simple relationship” because the proportionality symbol is commonly known.

• Before going into Results and discussion, the authors should provide algebraic details of how the correction factor is to be applied in order to account for a different medium (swimming/diving in water).

• Line 123-124: The authors need to more carefully phrase “wing-beat frequency is largely determined physiologically as the natural (resonance) frequency of the wing” because from recent literature we know that wingbeat frequency of many insects is supra-resonant. (For example, see Gau et al. 2022 “The hawkmoth wingbeat is not at resonance” and https://meetings.aps.org/Meeting/MAR24/Session/M38.1 ).

• The authors have put an appreciable effort into piecing together measurements from a variety of literature. I’d highly recommend that they make this data available in the supplementary material, in case they weren’t planning to already. For example, in the form of a spreadsheet.

7. PLOS authors have the option to publish the peer review history of their article (what does this mean?). If published, this will include your full peer review and any attached files.

Reviewer #2: **Yes: **Usama Bin Sikandar

---

## [Author Response · Author response to Decision Letter 1]

30 Apr 2024

We have uploaded a pdf containing response to all reviewer comments.

---

## [Editor Report · Decision Letter 2]

2 May 2024

Universal wing- and fin-beat frequency scaling

PONE-D-23-07976R2

Dear Dr. Hecksher,

We’re pleased to inform you that your manuscript has been judged scientifically suitable for publication and will be formally accepted for publication once it meets all outstanding technical requirements.

Kind regards,

Roi Gurka

Academic Editor

PLOS ONE
---

## [Editor Report · Acceptance letter]

10 May 2024

PONE-D-23-07976R2 

PLOS ONE

Dear Dr. Hecksher, 

I'm pleased to inform you that your manuscript has been deemed suitable for publication in PLOS ONE. Congratulations! Your manuscript is now being handed over to our production team.

Kind regards, 

on behalf of

Dr. Roi Gurka 

Academic Editor

PLOS ONE